# Long-Term Stable Complementary Electrochromic Device Based on WO_3_ Working Electrode and NiO-Pt Counter Electrode

**DOI:** 10.3390/membranes13060601

**Published:** 2023-06-15

**Authors:** Yajie Ke, Zitao Wang, Haiyi Xie, Mahmoud A. Khalifa, Jianming Zheng, Chunye Xu

**Affiliations:** 1Hefei National Research Center for Physical Sciences at the Microscale, Department of Polymer Science and Engineering, University of Science and Technology of China, Hefei 230026, China; keyajie@mail.ustc.edu.cn (Y.K.); wangzt@mail.ustc.edu.cn (Z.W.); xiehy@mail.ustc.edu.cn (H.X.); 2Anhui Province Key Laboratory of Condensed Matter Physics at Extreme Conditions, High Magnetic Field Laboratory, The Hefei Institutes of Physical Science, Chinese Academy of Sciences, Hefei 230031, China; khalifa@hmfl.ac.cn; 3Physics Department, Faculty of Science, Mansoura University, Mansoura 35516, Egypt

**Keywords:** complementary electrochromic devices, redox couple, tungsten oxide, nickel oxide, smart windows

## Abstract

Complementary electrochromic devices (ECDs) composed of WO_3_ and NiO electrodes have wide applications in smart windows. However, they have poor cycling stability due to ion-trapping and charge mismatch between electrodes, which limits their practical application. In this work, we introduce a partially covered counter electrode (CE) composed of NiO and Pt to achieve good stability and overcome the charge mismatch based on our structure of electrochromic electrode/Redox/catalytic counter electrode (ECM/Redox/CCE). The device is assembled using a NiO-Pt counter electrode with WO_3_ as the working electrode, and PC/LiClO_4_ containing a tetramethylthiourea/tetramethylformaminium disulfide (TMTU/TMFDS^2+^) redox couple as the electrolyte. The partially covered NiO-Pt CE-based ECD exhibits excellent EC performance, including a large optical modulation of 68.2% at 603 nm, rapid switching times of 5.3 s (coloring) and 12.8 s (bleaching), and a high coloration efficiency of 89.6 cm^2^·C^−1^. In addition, the ECD achieves a good stability of 10,000 cycles, which is promising for practical application. These findings suggest that the structure of ECC/Redox/CCE could overcome the charge mismatch problem. Moreover, Pt could enhance the Redox couple’s electrochemical activity for achieving high stability. This research provides a promising approach for the design of long-term stable complementary electrochromic devices.

## 1. Introduction

According to statistics, building energy accounts for about 45% of total societal energy consumption [1,2]. Windows transfer heat quickly, leading to high building energy costs; smart windows are a promising choice to address the energy consumption of windows [3]. Electrochromic glass is one of the types of energy-saving window glass; it can achieve active regulation of light transmission or absorption under the action of an electric field, thereby selectively absorbing or reflecting external heat radiation and preventing indoor heat loss to achieve the purpose of reducing temperature-control energy consumption [4,5,6]. A conventional electrochromic device (ECD) could be considered as a “sandwich” type electrochemical device, with five functional interlayers: a transparent conductive layer, an electrochromic (EC) layer, an ion conduction layer (electrolyte layer), an ion storage layer (counter electrode layer) and another transparent conductive layer [7]. Tungsten trioxide (WO_3_) has attracted tremendous attention as cathodic electrochromic material (ECM) owing to its exceptional properties, including low cost, high transmittance modulation, and favorable coloration efficiency [8,9]. The counter electrode (CE) layers matched with WO_3_ are usually anodic color-changing materials such as nickel oxide (NiO) [10,11], vanadium pentoxide (V_2_O_5_) [12] and Prussian blue (PB) [13]; taken together, these are known as complementary electrochromic devices (ECDs), with synergistically enhanced color-changing effects. However, the mismatch of charge capacity between cathode and anode films results in poor stability of these ECDs, as evidenced by the tendency toward uneven coloration and bubbles after multiple cycles [14,15,16,17]. To ensure the practical application of ECDs in smart windows, electronic displays and anti-glare mirrors, further improvement of the cycling performance and stability of the devices remains the main focus of current research [18,19,20].

Exploring the stability of electrochromic devices from the perspective of rational structural design means searching for device structures that could achieve stable electron and ion transport during electrochemical redox reactions. It has been shown that the addition of redox couples that spontaneously gain and lose electrons to the electrolyte can help balance the charge and thus drive the device’s operation [21,22,23]. For example, the electrochromic system designed by Georg et al. [21] does not contain an ion storage layer; it is replaced by a redox couple (I^−^/I_3_^−^) in the electrolyte and a thin platinum film on the transparent conductive glass. Kim et al. [24] introduced ferrocene (Fc) into the electrolyte layer as an anode species to improve the performance of WO_3_-based ECDs; however, the transmittance modulation ability was poor (only 52%) due to the absence of secondary EC films. An ideal EC window is transparent and of neutral color when in the bleached state (T_vis_ > 70%), while enabling a high optical modulation in the colored state, i.e., T_vis_ should be below 10% [25]. Therefore, it is still a general trend to develop complementary WO_3_-based ECD to obtain the capability of high modulation [26,27]. As the most representative electrochromic anode material, nickel oxide (NiO) has a stable optical modulation window in the visible region, low cost and easy preparation; thus, it has high research value and application prospects [26]. Diao et al. [28] revealed the electrochromic mechanism of WO_3_-NiO devices subjected to charge capacity and optical degradations by cyclic voltammetry. Ma et al. [29] reported 1D WO_3_/NiO nanorod-based ECDs with high performance and fast switching at a low applied potential; the devices only had 1000 cycles, which is insufficient for smart windows. Compared to the corrosive and naturally yellowish I^−^/I_3_^−^ and Fc, tetramethylthiourea/tetramethylformaminium disulfide (TMTU/TMFDS^2+^) redox couples are highly promising for ECDs due to advantages such as easy handling, colorlessness and lack of visible-light absorption [21,23]. TMTU/TMFDS^2+^ has been successfully introduced into ECDs by researchers; however, its reactivity is still insufficient relative to the high charge storage WO_3_ working electrode. In dye-sensitized solar cells (DSSCs), catalytic counter electrodes (CCEs) can significantly improve the electrochemical activity of redox couples [30,31,32]. The choice of CCE materials mainly includes precious metals, carbon materials, transition metal compounds and conductive polymers. Pt catalytic counter electrodes have received attention because of their good catalytic activity and transparency in the visible region [33,34,35]. In our previous research [36,37], we successfully developed partially covered CE for photoelectrochromic devices via the combination of DSSC and ECD. In this work, we aim to introduce the approach of partially covered counter electrodes (CEs) to develop a complementary electrochromic device.

In this research, a partially covered CE composed of NiO and Pt was applied to a complementary electrochromic device. The NiO anode electrochromic layer was prepared in the middle part of the conductive glass substrate; then, the Pt catalytic layer was introduced around it to form a bifunctional electrochromic counter electrode. The NiO thin film component endows the device with some advanced properties, such as large light modulation rate and fast switching speed. The catalytic properties of Pt allow the redox TMTU in the electrolyte to exhibit higher electrochemical activity, thus constructing a reversible redox system in WO_3_-based electrochromic devices and enabling long-term stable operation. This work provides a feasible idea for EC smart windows with low energy consumption and high stability.

## 2. Experimental Section

### 2.1. Materials

Pure tungsten powder (W, 200 mesh, 99%), hydrogen peroxide solution (H_2_O_2_, 30%), ethanol (99.8%), nickel (II) nitrate hexahydrate (Ni(NO_3_)_2_·6H_2_O, 98%), urea (CON_2_H_4_, 99%) and propylene carbonate (C_4_H_6_O_3_) were all obtained from Sinopharm Chemical Reagent Co., Ltd. (Shanghai, China). Tetramethylthiourea (TMTU, 98%), nitrosyl tetrafluoroborate (NOBF_4_, 95%) and lithium perchlorate (LiClO_4_, 99%, anhydrous) were purchased from Alfa Aesar (Shanghai, China). Pt-Catalyst T/SP was purchased from Solaronix (Aubonne, Switzerland). All solvents and chemicals were of analytical grade and used without further purification. De-ionized water with a resistivity of 18.2 MΩ·cm was obtained by filtration from the laboratory’s water purification system (Millipore S.A.S 6712, Molsheim, France). Both the electrochemically deposited WO_3_ films and the hydrothermal prepared NiO films were deposited and grown on the conducting transparent indium tin oxide (ITO)-coated glass substrate with a sheet resistance of 10 Ohm/sq.

### 2.2. Preparation of WO_3_ Films

WO_3_ electrochromic electrodes were prepared by electrodeposition using transparent homogeneous perovskite solution in our previous work via two steps, as follows [38].

The first step was the preparation of WO_3_ precursor solution: 6 g W powder and 60 mL H_2_O_2_ were deposited into a clean beaker and stirred; the mixing process was accompanied by a violent reaction. The above reaction solution was filtered twice to remove the unreacted W powder, and a colorless translucent solution was obtained. The solution was transferred to an oil bath and refluxed continuously for 12 h at 51 °C, 2 h at 65 °C and 30 min at 85 °C. When the color of the solution turned pure yellow, 60 mL of ethanol was added immediately, and the PPTA solution was obtained after the final reflux at 50 °C for 24 h, which was stored in the refrigerator (4 °C) and sealed for later use.

The second step was the preparation of the WO_3_ electrochromic electrode by electrodeposition using a three-electrode system (CHI-660D, CH Instruments, Shanghai, China). The PPTA solution obtained in the previous step was used as the electrolyte, with platinum (Pt) sheet and silver wire as counter electrode and reference electrode, respectively. The ITO (2.5 cm × 5 cm) was used as the working electrode. By applying a constant voltage of 0.56 V (vs. Ag/Ag^+^) for 150 s, blue WO_3_ film could be observed. The films were immersed in ethanol for one minute, dried naturally, and then annealed at 300 °C for 30 min in a muffle furnace to obtain the transparent and uniform crack-free WO_3_ electrochromic electrodes.

### 2.3. Preparation for NiO, Pt and NiO-Pt Composite Films

Preparation of NiO electrode by hydrothermal treatment: nanostructured NiO films were grown directly on ITO-coated glass through hydrothermal and annealing treatment. The ITO substrates were immersed in de-ionized water, acetone and ethanol for ultrasonic washing and then dried for use. Briefly, urea (0.2326 g) and Ni(NO_3_)_2_·6H_2_O (0.0961 g) were dispersed in 40 mL of de-ionized water. After stirring for 30 min, the obtained green transparent solution was transferred to a 50 mL Teflon-lined stainless-steel autoclave. Then ITO-coated glass was inserted and tilted into the reaction solution, and the autoclave was sealed and maintained at 180 °C for 1.5 h. After natural cooling to room temperature, the ITO glass was taken out and washed with de-ionized water. Finally, the drying film was annealed at 300 °C in air for 1 h to obtain NiO film.

Preparation of the Pt electrode by annealing treatment: an art brush was dipped into the Pt-Catalyst T/SP chemical reagent, and the reagent was brushed onto the ITO-coated glass substrate as evenly as possible. Then, it was placed on the heating plate to gradually warm up to 300 °C and maintained at this temperature for 1 h to obtain the Pt electrode.

Preparation of NiO-Pt composite electrode: the composite film was prepared by hydrothermally growing NiO film in the middle of the ITO-coated glass and then brushing the Pt catalyst around the substrate. We used a certain width of polyimide tape to cover the outer ring of ITO glass to ensure that no NiO film grew around the substrate during the hydrothermal process. Other experimental details are consistent with the preparation process of the single electrode described above.

### 2.4. Preparation of Electrolyte

We mixed 2.1278 g LiClO_4_ and 1.3223 g TMTU in 20 mL PC under stirring to obtain a light yellow solution. Then, 0.2336 g NOBF_4_ was added to the solution and stirred vigorously until dissolved [21]. All the devices and related analytical tests were executed with this electrolyte unless otherwise stated.

### 2.5. Assembly of Electrochromic Devices

We assembled three types of devices, as shown in Figure 1. These devices were prepared with the same WO_3_ working electrodes and electrolytes but employing different counter electrodes, including NiO, Pt, and NiO-Pt films, which were labeled as W-N, W-P, and W-NP. Ultraviolet (UV) curing sealant was used to bond the edge of the electrodes with a space of about 150 μm. The liquid electrolyte was injected into the gap between the two electrodes by syringe; then, the device was sealed and cured. The active area of the device was the part enclosed by the adhesive, with an area of 2.3 × 4 cm^2^.

### 2.6. Characterization

The morphology and thickness of the films were observed using field emission scanning electron microscopy (SEM, SU8220, Hitachi, Japan). The crystalline structures of the films were characterized by X-ray diffraction (XRD) analysis (X’Pert PRO; PANalytical B.V., Almelo, The Netherlands), in which Cu-Kα was used as the X-ray source. The chemical composition analysis of the films was performed by X-ray photoelectron spectroscopy (XPS, ESCALAB 250, Thermo-VG Scientific, East Grinstead, West Sussex, UK), and all of the XPS binding energies were calibrated using the contaminant carbon (C 1 s) as a reference. The electrochemical and optical properties of the ECDs were investigated and analyzed using an electrochemical analyzer (CHI-660D, CH Instruments, Shanghai, China) and UV-vis-NIR (near-infrared) spectroscopy (V-670, Jasco, Tokyo, Japan).

## 3. Results and Discussion

### 3.1. Structure and Morphology of Thin Films

X-ray diffraction (XRD) patterns were obtained to evaluate the crystal structure of the synthesized WO_3_ and NiO films. In Figure 2a, a broad peak was observed in the range of 20–30°, which verified that the as-deposited WO_3_ film possess an amorphous characteristic and highly disordered structure, consistent with a previous report [32]. Removing the influence of the ITO glass crystal form, the three diffraction peaks (2θ = 37.4, 43.3 and 62.9°) of NiO can be indexed to the (111), (200) and (220) planes of the cubic NiO phase (JCPDS 47-1049). In addition, there are no excess peaks, revealing that the fabricated NiO films are cubic crystal phase. X-ray photoelectron spectroscopy (XPS) was further conducted to assess the surface chemical composition and oxidation state of the WO_3_ and NiO films. Only W, Ni, O and C peaks existed in the XPS survey spectra, with the carbon coming from the background. The spin-orbit doublets in the high-resolution W 4f spectrum correspond to W 4f_7/2_, W 4f_5/2_ and W 5p_3/2_ peaks, which are located at 35.9, 38.1 and 41.9 eV, respectively (Figure 2b). The tungsten is composed of W^5+^ and W^6+^ valence states. The main bimodal peak of W^6+^ is 38.2 eV/36.1 eV, and the weak bimodal peak of 36.9 eV/34.7 eV is attributed to W^5+^, indicating the existence of oxygen vacancies in the sample [39]. The oxygen vacancies are helpful for improving the conductivity of WO_3_ films and the dynamics of ion intercalation/deintercalation behavior [40]. As shown in Figure 2c, the high-resolution XPS spectrum of Ni 2p includes photoelectron peaks (850 to 858 eV) and satellite peaks (858 to 870 eV) caused by the shake-up processes. The peaks with binding energy of 853.7 eV and 855.8 eV are attributed to Ni^2+^ (NiO) and Ni^3+^ (Ni_2_O_3_), respectively. The satellite peaks do not contain any other additional information on surface chemistry.

Figure 3a shows the typical SEM image of WO_3_ with a thickness of ~575 nm deposited on the ITO substrate; the film is relatively homogeneous and dense, which helps to avoid volume expansion after ion embedding in the film. As exhibited in Figure 3f, the morphology of NiO consists of interleaved growth of nanosheets, forming a porous interconnecting network structure with a large specific surface area. The porous structure could accelerate the interface electron kinetics between the film and electrolyte and significantly improve the electrochromic performance. Figure 3e shows the microscopic morphology of the junction of NiO-Pt composite film; NiO and Pt are distributed regionally in the counter electrode. Pt in the ITO substrate indicates an island-like distribution without specific morphology and thickness.

### 3.2. The Complementary Effects of WO_3_ and NiO Electrochromic Processes

In order to better demonstrate the complementary spectral matching of WO_3_ and NiO, the transmittance spectra were measured over the wavelength region from 300 to 800 nm for WO_3_ and NiO films on ITO-coated glass in the colored and bleached states (Figure 4a,d). The pure WO_3_ film could achieve a large optical modulation (Δ*T* = *T*_b_ − *T*_c_, where *T*_b_ and *T*_c_ denote the transmittance of the sample at the bleached and colored states, respectively) during the coloring process, especially at 650 nm, where Δ*T* can be as high as 85.7%. In fact, the transmission rate of the ITO conductive glass substrate used in the experiment is about 90%, which shows that the WO_3_ film in the bleached state has excellent transmittance for visible light. It is noteworthy that WO_3_ film in the colored state has a transmission peak of about 40% at 400 nm, which indicates that its spectral absorption performance is slightly weak in the region of 300–500 nm. The original NiO film is light yellow in the visible region, while the transmittance decreases uniformly by about 40% after the coloring process. WO_3_ and NiO are materials with opposite color-changing properties, which can be further assembled into electrochromic devices. Referring to the illustration in the transmission spectrum, when WO_3_ gains electrons and undergoes a reduction reaction to turn blue, NiO as the counter electrode loses electrons and undergoes an oxidation reaction to turn brown-black, which could strengthen the light absorption of the whole device and lead to lower spectral transmittance, especially at 400 nm. When WO_3_ is oxidized to a transparent state, NiO is also reduced to light-yellow state, and the whole device returns to the faded state. In conclusion, the color change of WO_3_ and NiO films corresponds to different wavelength ranges in the visible region; thus. the device assembled from both can realize a synergistic electrochromic effect, which is expected to realize dark colored states and high absorption in the visible spectrum region.

One of the most important criteria for selecting an electrochromic material is its coloration efficiency (*CE*). This refers to the ratio of the change in optical density (*OD*) to the per unit charge density (Q/A) inserted into or extracted from the electrochromic material. The specific calculation of *CE* is shown in the following equation.
$$CE=△OD/Q=log(T_{b}/T_{c})/Q$$

Figure 4 shows the curves of the Δ*OD* of WO_3_ and NiO at 400 nm and 650 nm, respectively, as the charge density increases. The slope value obtained from the fit of the straight line part at the front of the curve is the *CE* value. When Pt sheets are used as both reference and counter electrodes in the two-electrode system, observation of the values of the charge in the transverse coordinate shows that the charge required to apply −1.7 V for the WO_3_ film is 35–40 mC·cm^−2^, while the charge required to apply 1.5 V for the NiO film is 10–12 mC·cm^−2^, which indicates that the electrochemical reactivities of WO_3_ and NiO are not comparable. This further supports that the conventional WO_3_-NiO electrochromic devices have difficulty achieving long-term stable operation. According to Figure 4b, the *CE* of WO_3_ at 400 nm is only 11.6 cm^2^·C^−1^. When WO_3_ and NiO are assembled into a device, the theoretical *CE* of the device could be significantly improved. The higher *CE* means that the complementary devices can achieve a faster ion embedding rate during the electrochromic coloring process with the same applied voltage drive, thus realizing a fast color change.

### 3.3. Structure Configuration and Working Mechanism of the ECDs

To investigate the effect of the NiO-Pt composite counter electrode on the ECDs’ performance, two other conventional devices with NiO and Pt as counter electrodes were prepared for comparison. Figure 1a–c shows the schematic structure of the three devices, which are named W-N, W-P and W-NP for the convenience of presentation. “W”, “N” and “P” represent the WO_3_ working electrode and the NiO and Pt counter electrodes, respectively.

W-N is a common thin-film cell-type sandwich-type device structure: ITO/WO_3_/LiClO_4_-PC/NiO/ITO. Figure 1a illustrates the working mechanism of W-N during the coloring process. By applying a negative bias to the WO_3_ working electrode, electrons are injected into the WO_3_ film through an external circuit; in order to maintain charge balance, an equal amount of Li ions from the electrolyte migrates to insert into the WO_3_ film. The underlying physics involved in the electrochromic reaction can be represented using the following redox equations:(1)$$\mathrm{Cathode}:{WO}_{3}(bleached)+x{Li}^{+}+xe^{-}⇋{Li}_{x}{WO}_{3}(colored)$$

The charge transfer between W^6+^ and W^5+^ leads to a change in the photo-effected WO_3_, accompanied by the film switching from transparent to blue. The electrochromism of NiO films in the LiClO_4_/PC electrolyte is caused by the continuous reversible formation of dark-brown LiyNiOx during Li ion embedding/extraction, which can be expressed by the redox reactions of Equations (2) and (3).
(2)$$Anode:{NiO}_{x}+y{Li}^{+}+ye^{-}\to {Li}_{y}{NiO}_{x}$$
(3)$${Li}_{y}{NiO}_{x}(bleached)⇋{Li}_{\left(y-z\right)}{NiO}_{x}(colored)+z{Li}^{+}+ze^{-}$$

For the W-P and W-NP devices, Pt catalytic counter electrodes are introduced, and TMTU/TMFDS^2+^ redox couples are added to the electrolyte. As shown in Figure 1, when the WO_3_ film is connected to the negative terminal of the power supply, TMTU releases electrons at the Pt electrode to be oxidized to TMFDS^2+^, thus ensuring charge balance to maintain the coloration state. Equation (4) represents the conversion between TMTU and TMFDS^2+^ in the redox reaction.
(4)$$2TMTU-2e^{-}⇋{TMFDS}^{2+}$$

### 3.4. Electrochemical Properties of Different Counter Electrodes

Electrochemical impedance spectroscopy (EIS) measurements were employed to investigate the intrinsic kinetics of the NiO, Pt and NiO-Pt electrodes. Figure 5 compares the Nyquist plots of symmetrical cells assembled with different counter electrodes. Impedance spectra were measured from 10^−2^ to 10^5^ Hz, with 5 mV of amplitude and a zero bias of potential. The intercept of Nyquist plots on the real axis reflects series resistance (*R_s_*), and the semicircle in the middle frequency region indicates the value of the charge-transfer resistance (*R_ct_*) at the electrode/electrolyte interface. In general, the gain or loss of electrons occurs where the electrode is in contact with the electrolyte; we focus on comparing the charge-transfer resistance (*R_ct_*). Compared to that of the NiO electrode, the *R_ct_* value of the composite NiO-Pt electrode partially covered with Pt was significantly decreased. This indicates that the reactivity of partly covered NiO-Pt CE is enhanced in the presence of Pt; the electrons could be flowed through the external circuit to the counter electrode easily.

### 3.5. Electrochromic Performance of ECDs

Light modulation range is an important characteristic of electrochromic devices. Figure 6 shows the UV-visible transmittance curves of W-N, W-P and W-NP devices in the colored state (−1.5 V, 40 s) and bleached state (1.2 V, 40 s). The measured wavelength range is 300–800 nm. As indicated in Figure 6a, the light modulation range is Δ*T* = 59.2% at 600 nm for W-P, while W-N and W-NP could reach a ΔT of 68% at the same wavelength. The increase of the light modulation range indicates better electrochromic performance, mainly because NiO, as an anode electrochromic material, has the inherent advantage of a large light modulation range. Moreover, the working electrode and counter electrode of the device undergo the color change process simultaneously, thus reflecting a larger light modulation range than the monolayer WO_3_ film. Comparing the physical diagrams of the coloring states and optical transmittance of the three devices, we found that W-P can only switch between colorless transparent and blue because only WO_3_ films are involved in the color change; the transmittance at 400 nm is as high as 40%. The coloration state of W-N and W-NP can reach blue-violet color, and the peak in the colored state drops to about 15%.

The significant impact of different counter electrodes on the performance of ECDs is also reflected in the switching time and coloration efficiency. The electrochromic switching time was characterized by the chronoamperometry method and in situ optical transmittance change, and a square-wave voltage signal with a range of −1.5 V and 1.2 V with a pulse width of 40 s was applied to measure the real-time transmittance change of different ECDs at 600 nm. The response time is defined as the time required for a 90% change in transmittance. According to Figure 7a, the coloring time (*t*_c_) and bleaching time (*t*_b_) of W-N are 5.6 s and 6.5 s, respectively. In comparison, *t*_c_ (W-NP) = 12.8 s and *t*_b_ (W-NP) = 5.3 s; the increase in coloring time is attributed to the higher surface reactivity of the Pt catalytic material at the periphery of the composite electrodes in the coloring process, which preferentially participates in the electrode reaction. Thus, we could observe a coloring phenomenon similar to the diffusion from the outer ring to the inner one, accompanied by a deepening of the coloring state. Figure 7b reveals that the coloring efficiency of W-N and W-NP is 91.2 cm^2^·C^−1^ and 79.7 cm^2^·C^−1^, respectively. There is a significant difference between the above two devices; the CE of the W-P device is only 41.2 cm^2^·C^−1^, which is similar to that of the WO_3_ film. The high CE implies low energy consumption, indicating that the composite electrode complementary electrochromic devices has great potential for environmentally friendly applications.

### 3.6. Cycle Stability of the ECDs

Long-term cycle stability is an important factor for practical applications of ECDs. The change of charge number in the switching process could be effectively reflected by a double-step chronoamperometry method, which could also be used to evaluate and compare the cycling ability of W-N and W-NP devices. The cycling stability of our devices was carried out by switching the applied coloring (−1.5 V)/bleaching (1.2 V) voltage in 20 s intervals to ensure that the device transmittance can be significantly changed and cycled for multiple cycles. In Figure 8a, the W-N device exhibits significant current density degradation after only 100 cycles, accompanied by uneven coloring. From Figure 8b, it can be found that after 1000 cycles, the spectrum of the W-N device in both the colored and bleached states declines, and the transmittance modulation ability decreases. This situation originates from the low charge capacity and poor cycling stability of the NiO film itself, making it difficult to balance the charge required for the WO_3_ working electrode. As for the W-NP device, the current density–time curve stays at a similar level during 10,000 cycles, demonstrating excellent stability. Figure 8c shows the spectral modulation curves of the W-NP device after different cycles; the transmittance modulation at 650 nm is labeled. After 5000 and 10,000 cycles, the transmittance modulation of W-NP ECD is reduced by 4.88% and 12.34%, respectively. It is worth noting that as the number of cycles increases, the transmittance of the device’s coloring state at 400 nm also gradually increases but still maintains a deeper coloring state. The above phenomenon indicates that the NiO film component in the composite counter electrode is only degraded and does not completely fail, which proves that the ECM/Redox/CCE structure greatly moderates the damage of the NiO material by the electrode process and further improves the device stability and durability. In summary, the electrochromic performance of the partly covered NiO-Pt CE-based ECD is better than other types of devices; the relevant data are summarized in Table 1.

## 4. Conclusions

In conclusion, the ECM/REDOX/CCE structure is introduced into the complementary electrochromic system, and a bifunctional NiO-Pt CE with electrochromic and electrocatalytic activities was prepared. The corresponding new ECD shows significant electrochromic performance improvement, with a transmittance contrast of 67.3% at 600 nm and fast response times. During the durability characterization, the transmittance change of ECD remained at 54% after 10,000 cycles, which was about 91% of the original state. The improved EC performance of the device is attributed to the synergy of the reaction process of the WO_3_ and NiO electrodes, and the problem of electrode charge mismatch is overcome via the introduction of a partially covered counter electrode of NiO-Pt. This research provides promising strategies for the design of complementary ECDs, which can be generalized and applied to smart windows in the future.

## Figures and Tables

**Figure 1 membranes-13-00601-f001:**
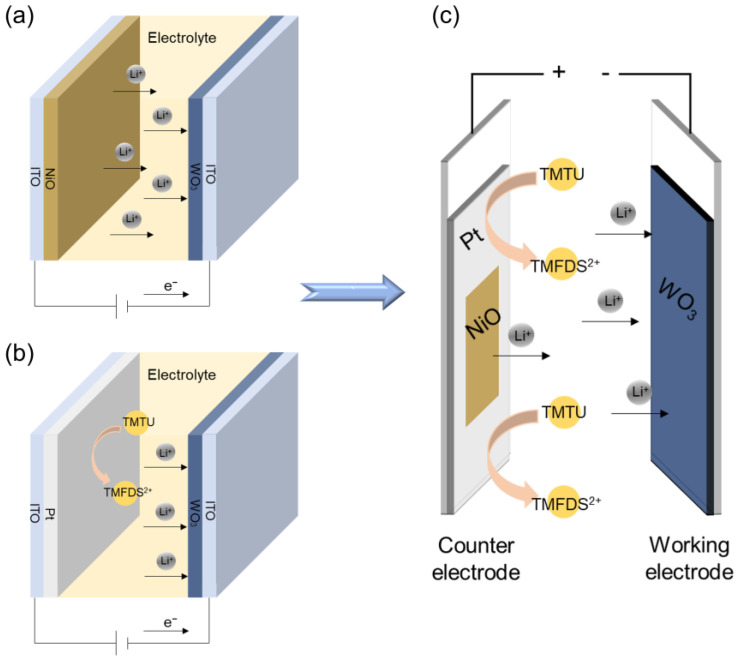
Structures of different devices: (**a**) conventional ECD (W-N), (**b**) ECD with ECM/Redox/CCE structure (W-P) and (**c**) ECD with hybrid counter electrode (W-NP).

**Figure 2 membranes-13-00601-f002:**
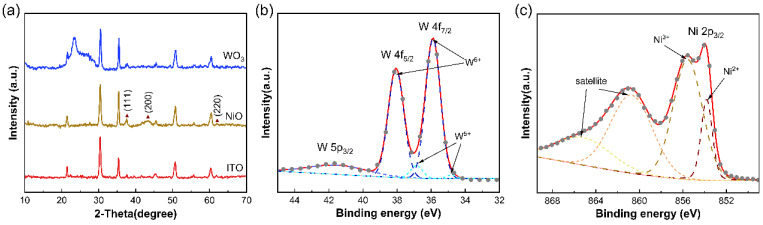
Characterizations of WO_3_ ECE and NiO CEE. (**a**) XRD patterns of ITO/glass, the as-prepared WO_3_ film and NiO film. (**b**,**c**) High-resolution XPS spectra of W 4f in WO_3_ and Ni 2p in NiO, respectively.

**Figure 3 membranes-13-00601-f003:**
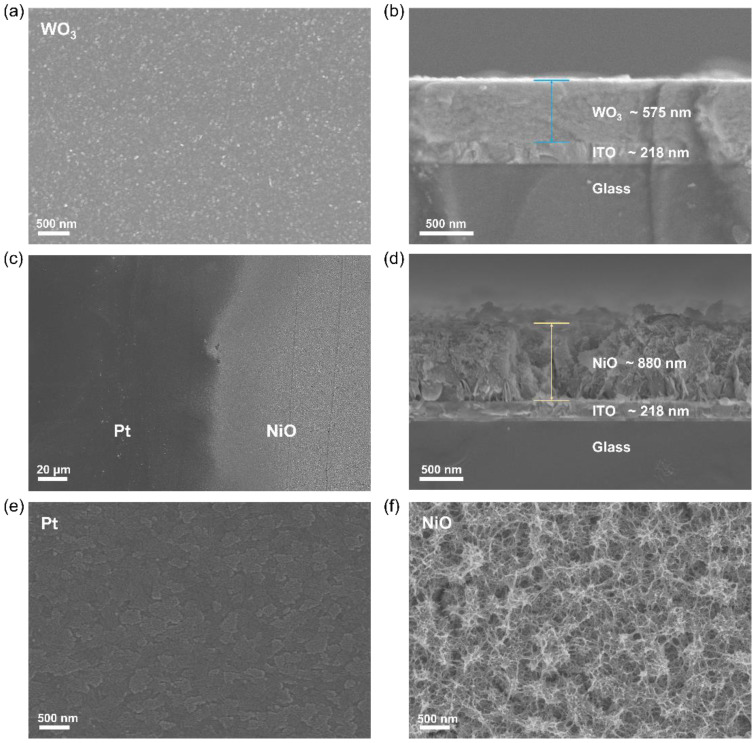
SEM images of (**a**) WO_3_ film, (**c**) NiO-Pt film, (**e**) Pt film and (**f**) NiO film. Cross-sectional SEM images of (**b**) WO_3_ and (**d**) NiO electrodes.

**Figure 4 membranes-13-00601-f004:**
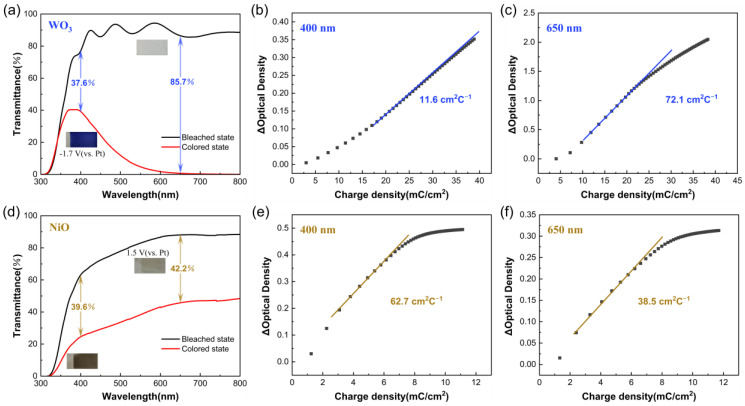
UV-vis transmittance curves of (**a**) as-prepared WO_3_ films and (**d**) NiO films in the wavelength range of 300–800 nm (color figure online). The optical density (*OD*) with respect to the charge density for (**b**,**c**) WO_3_ films and (**e**,**f**) NiO films at 400 nm and 600 nm.

**Figure 5 membranes-13-00601-f005:**
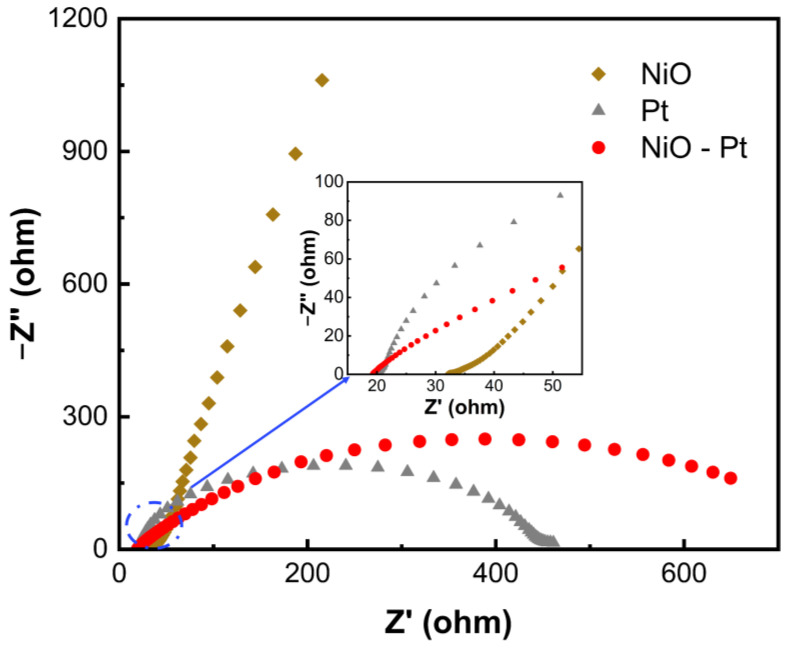
Nyquist plots of symmetric cells assembled from NiO, Pt and NiO-Pt counter electrodes (illustration is an enlarged view of the selected region).

**Figure 6 membranes-13-00601-f006:**
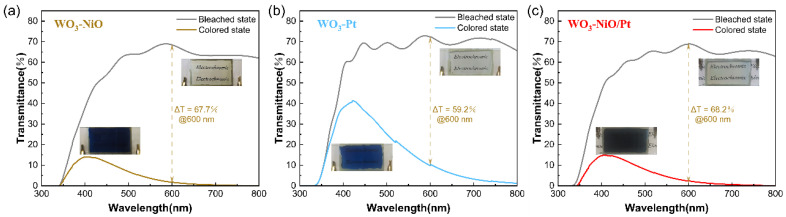
UV-vis transmittance curves of electrochromic devices (**a**) W-N, (**b**) W-P and (**c**) W-NP measured at 1.2 V (bleached state) and −1.7 V (coloring state).

**Figure 7 membranes-13-00601-f007:**
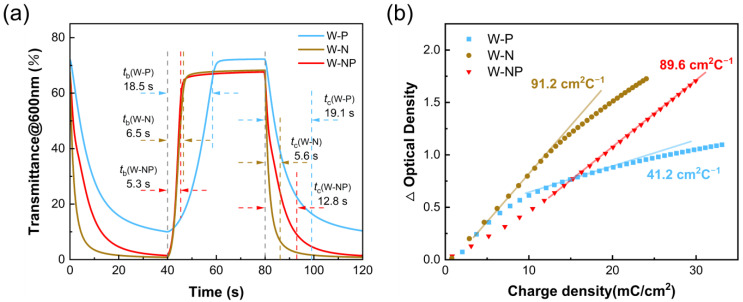
(**a**) Transmittance switching time characteristics between the bleached and colored states for electrochromic devices measured 1.2 and −1.5 V at 600 nm. (**b**) Variations of *OD* with respect to the charge density for W-P, W-N and W-NP.

**Figure 8 membranes-13-00601-f008:**
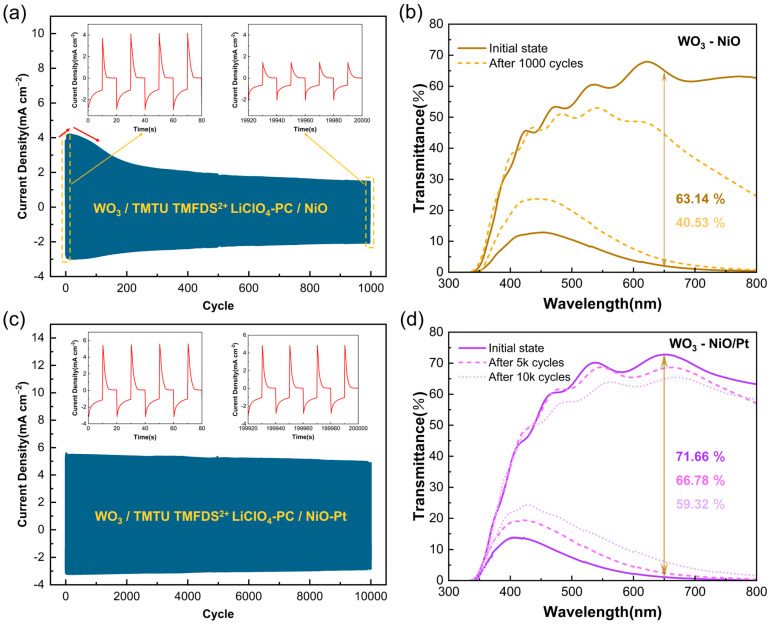
(**a**) W-N device was measured with multi-potential steps between −1.5 V (20 s) and 1.2 V (20 s) for 1000 cycles. (**b**) Transmittance spectra of W-N initially and after 1000 cycles. (**c**) W-NP device measured with multi-potential steps between −1.5 V (20 s) and 1.2 V (20 s) for 10,000 cycles. (**d**) Transmittance spectra of W-NP initially and after 5 K and 10 K cycles; 1 K = 1000.

**Table 1 membranes-13-00601-t001:** Summary of electrochromic properties for W-N, W-P and W-NP.

ECDs	Transmittance Modulation @ 600 nm (%)	t_c_ (s)	t_b_ (s)	Coloration Efficiency (cm^2^·C^−1^)
W-N	67.7	6.5	5.6	91.2
W-P	59.2	18.5	19.1	41.2
W-NP	68.2	5.3	12.8	89.6

## Data Availability

Data sharing is not applicable to this article.

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
