# Peer review of "Long-Term Stable Complementary Electrochromic Device Based on WO3 Working Electrode and NiO-Pt Counter Electrode"

_membranes, 2023, doi:10.3390/membranes13060601_

Round 1
Reviewer 1 Report
Comments and Suggestions for Authors
Decision:
Major Revision
Comments
The authors reported a Long-term-stable complementary electrochromic device base on WO3 and NiO/Pt composite counter electrodes. The author showed high optical modulation of 68.2 % which is a good result here however there are several drawbacks in the manuscript. Articles have huge similarities index with other published papers. I have the following points outlined below to improve the scientific quality.
- There is a minor correction in the title( its Based on WO3 on WO3 and NiO/Pt composite)
- Reduce the similarities in the paper and remove fully copied sentences from other articles.
- In the abstract section, the author should highlight the other important results with exact values achieved.
- In the introduction, section author should relate the work with previous research and highlight the importance of Wo3 and NiO in other optical devices with more references.
https://www.nature.com/articles/s41598-020-65191-x
https://iopscience.iop.org/article/10.7567/JJAP.52.03BB08/meta
https://www.nature.com/articles/s41598-023-28356-y
https://www.mdpi.com/1996-1944/15/19/6658
4. In the experimental section author should add device structures images or flow charts and explain the steps clearly.
5. Figure 1 should be divided into 2 figures (separate XRD and SEM Images.
6. Remove the inset of SEM images. (Structural image)
7. Figure 3 is cited first in the text and then in Figure 1.
8. Move Figure 3 in the experimental part ( add with comments 4)
9. Author should add the thickness of these layers. (Wo3, NiO, TiO)
10. SEM dates need more explanation and the exact difference between Figure 1(d) and 1(e).
11. Figure 1(e) shows NiO on ITO. What's the thickness of NiO here?
12. What about the SEM images of the NiO/Pt composite?
13. In Results I suggest adding a Table for all the 3 devices and adding each result for comparison. Highlighting the best devices.
14. There are several grammatical errors. A proofread is required.
Comments on the Quality of English Language
There are several grammatical errors. A proofread is required and reduce the similarities index
Reviewer 2 Report
Comments and Suggestions for Authors
This is an interesting work, in particular based on novel device architecture using redox couple in the electrolyte and composite Pt/NiO electrode worthwhile to be published after some major revisions.
Some of the films characterizations need better description.
The authors mention an amorphous character for WO3 while the XRD pattern presents some peaks indexed (110, 002,..). Can the authors clarify this point ?
As compared to some literature, NiO appears poorly crystallized. Any comments ?
XPS : Fig. 1b, what is the meaning of pointing out the W5+ signature ? The sentence ..The tungsten element consists of..What is the proof of existing 5+ oxidation state as well as oxygen vacancies ?
The description of the XPS spectra needs to be completed.
Line 225 : Can the authors clarify the mention of the dual electrochromic effect ? Usually the dual band states for activities in both the visible and the NIR regions ?
Fig. 2 : caption chaege to be replaced by charge..why for NiO there is a plateau at high charge density and the evolution is rather straight for WO3 ?
Line 246 : Figure 4b to be replaced by Figure 2b
The sentence, The original NiO Film is not completely transparent..needs to be rewritten or at least to be more precise..Why do the transmittance curve of WO3 show transmittance and not the one of NiO ? The complementary in between NiO and WO3 is not new. Some comment scan be simplified. Or some references added.
Some references are missing. See for instance Dong, Diao et al., J of Materials Chem. C, 2018, Wang, Diao et al., SOLMAT, 2021
Line 268 : What is the meaning of photo-effected of WO3 ?
The long term stability shows quite a large resistance in the current density vs time ; Overall, the complete device suffers from a limitation in kinetics. Can the authors comment this point ?
Minor comments :
Introduction, line 44 : alum pentoxide to be replaced by vanadium pentoxide
Line 105 : 2.2 preparation
Reviewer 3 Report
Comments and Suggestions for Authors
This research is an experimental study aimed at improving the long-term stability of ECD by utilizing spontaneously redox reactions to control charge transfer between the cathode and anode electrode, which is a critical factor affecting durability.
And This paper compares the transmittance In Visible region wavelength, Charge Density, Long-term stability between a conventional ECD made with a general WO3 (tungsten oxide) cathode and NiO (nickel oxide) anode, and an ECD made with WO3 cathode and a hybrid electrode of NiO and PT (platinum) as the anode.
Comment 1)
I found the wrong word or ambiguous word. I wrote a few. English words and sentences need to be reviewed in detail.
‘ferropene’ -> what is that thing? Is it ferrocene? (line 59)
‘Materils’ -> ‘Materials’ (line 92)
‘Prepration’ -> ‘Preparation’ (line 105)
XRD graph – (2 2 1)plane -> I didn’t found that plane in JCPDS 47 – 1049. Maybe it is (2 2 0 ) Plane
‘W’/’T’/’C’ -> That letters mean Tungsten / Nickel / PT respectively. Is it correct?(line 259~260)
Comment 2)
l Page 1),line 43~44
There are several examples used in anode materials. Nickel oxide (NiO), aluminum pentoxide (V2O5) and Prussian blue. When checking NiO charge density in the paper, it is not enough to accommodate the capacity of the anode.
Are there not enough charge density for other anode materials like nickel oxide? (Ex; Add relevant references.)
Comment 3)
l Page 3), Section 2.2 to 2.3
In the case of tungsten films, electrical deposition was used, and in the case of Nickel, a hydrothermal deposition method using autoclave was used, is there any reason for deposition that differently?
And when bush Pt reagent on the NiO glass, did you check thickness or other things?
(ex: SEM data of Pt as shown in Figure1.)
Comment 4)
l Page 4) line 149
When you prepare electrolyte, you added 0.2336 g NOBF4 to the solution and stirred it.
Please explain the purpose and reason for adding NOBF4. And I also need a reference.
Comment 5)
l Page 4) Section 3.1 & Page 5) Figure 1
When looking at the XRD graph, nickel oxide has a cubic structure. Does this structure have a positive effect on the long-term stability?
When looking at the XPS graph, the intensity of nickel oxide between the oxidation and reduction states is relatively low. Is this effect due to the fact that nickel has a cubic structure or other thing?
Please explain the structure of NiO in detail.
Comment 6)
l Page 5) Figure 2
The platinum(PT) is mentioned as an additive for charge capacity through spontaneous redox reactions.
Do you have data on the cyclic voltammetry experiments for platinum, nickel oxide, and tungsten counter electrodes?
(Enter the correct reference and count electrodes for CV with Ag/Agcl)
Comment 7)
l Page 6) line 239~240
Why was a platinum electrode used as the reference and counter electrode spontaneously? Was the experiment conducted with a two-electrode system? Also, was the platinum electrode used as the electrode deposited on ITO or not?
(CV data of the three-electrode system is required with Ag/AgCl)
Comment 8)
l Page 6) Figure 2
Why are only the coloring voltages indicated and not the bleaching voltages? Is there any specific reason for this?
In this experiment about half cell, the electrolyte with TMTU was not used. Doesn't this have any effect on the results?
Comment 9)
l Page 8) Figure 5
The pictures are too small to confirm the results easily. It would be better to have larger pictures. It would also be helpful to attach separate experimental videos (Supplementary data)
Comment 10)
In repetitive experiments, oxidation persistence of PT is important.
Is there a way to check the degree of corrosion?
Comment 11)
l Page 10) Figure 7
It would be helpful to have a graph showing the change in transmittance as well as the graph showing the change in current density in Figure 7.
(As showing W-NP device (b)-(c), in the W-N device, I want show the transmittance data.)
Round 2
Reviewer 1 Report
Comments and Suggestions for Authors
The author addressed each of my comments and revised the manuscript very well. Therefore I accept this paper for publication in its current form.
Reviewer 2 Report
Comments and Suggestions for Authors
The revised version is acceptable in its present form. The authors have well integrated the comments and modifications.